# Trends in Tobacco Use among 9th Graders in Sweden, 1991–2020

**DOI:** 10.3390/ijerph20075262

**Published:** 2023-03-25

**Authors:** Jonas Raninen, Isabella Gripe, Martina Zetterqvist, Mats Ramstedt

**Affiliations:** 1Swedish Council for Information on Alcohol and Other Drugs (CAN), 116 64 Stockholm, Sweden; jonas.raninen@ki.se (J.R.); isabella.gripe@ki.se (I.G.); martina.zetterqvist@can.se (M.Z.); 2Department of Clinical Neuroscience, Karolinska Institutet, 171 77 Stockholm, Sweden; 3Centre for Alcohol Policy Research, La Trobe University, Melbourne, VIC 3086, Australia; 4Department of Public Health, Stockholm University, 106 91 Stockholm, Sweden

**Keywords:** tobacco, youth, survey, Sweden, snus, cigarettes

## Abstract

Tobacco use was measured with self-reports of lifetime use of cigarettes and snus to examine trends in tobacco use among Swedish 9th graders over the period 1991–2020. Annual school surveys with nationally representative samples of 9th-grade students in Sweden covering the period 1991–2020 with a total sample of 163,617 students. We distinguished between the use of cigarettes only, use of snus only, dual use (use of both cigarettes and snus), and total tobacco use (use of any of these tobacco products). In addition to a graphical description of trends in the various measures of tobacco use, the correlation between these trends was calculated with the Pearson correlation coefficient (Rxy). The prevalence of total tobacco use declined from 72% in 1991 to 36% in 2020. The declining trend in total tobacco use was positively correlated with the trend in dual use (Rxy = 0.98) and the trend in cigarette use only (Rxy = 0.87). The trend in total tobacco use was, on the other hand, negatively correlated with snus use only (Rxy = −0.41), and snus use only was negatively correlated with cigarette use only (Rxy = −0.71). The situation became different after 2017 when total tobacco use increased as a result of an increasing prevalence of snus use. The sharp decline in tobacco use among 9th graders in Sweden over the past three decades is driven by declining cigarette use. The correlations between the various forms of tobacco use suggest that snus use may have contributed to the decline in cigarette use and, by that, overall tobacco use. The situation changed after 2017 when a sharp rise in snus use seems to have increased total tobacco use among adolescents in Sweden. A possible explanation behind this development is the introduction of a new form of snus called “All white snus”, which was introduced in Sweden in 2014.

## 1. Introduction

Tobacco use is estimated to cause over 7 million deaths each year [1] and is a global health priority. Cigarette smoking is responsible for the major part of tobacco-related mortality, whereas smokeless tobacco (snus) at present is found to be associated with much lower risks for adverse health effects [2]. In international comparison, Sweden enjoys a low prevalence of smoking, especially among men. Sweden is among the countries with the lowest prevalence of current users, together with countries such as Australia, Canada, and India, which all have a prevalence of current tobacco smoking of below 20 percent [3]. Among young people, the prevalence of lifetime use of cigarettes in Sweden is among the lowest in Europe for both boys and girls [4]. This is thought to be partly because of the prevalence of smokeless tobacco, snus [5,6]. Still, there is a shortage of studies addressing trends in tobacco use among adolescents in Sweden and especially the interaction between the use of cigarettes, snus, and use of both cigarettes and snus, i.e., dual tobacco use.

In the adult population, the prevalence of dual users is low [7], whilst, among younger people, dual use is more common [8,9]. Dual use in adolescence is also found to be associated with a higher risk for dependence [10,11]. However, dual use could also be an indication of an ongoing switch from cigarettes to snus with possible positive effects in terms of declining smoking. The importance of smokeless tobacco products as a way to reduce harm from smoking has been debated internationally [12], with a positive net effect of snus achieved only if those using snus are switching from smoking cigarettes [2]. By analyzing long-term trends in various forms of tobacco use, it is possible to gain new insights into this matter by elucidating how they are related across time.

The overall aim of the present study is to examine trends in tobacco use among Swedish adolescents between 1991 and 2020. More specifically, we will:

1. Describe trends in the prevalence of tobacco use among Swedish 9th graders overall and separately for snus solely, cigarette smoking solely, and dual use of both snus and cigarettes; 2. Assess to what extent these trends are correlated across the period 1991–2020.

Among young people, the prevalence of daily smoking is usually higher among females [4], whilst the prevalence of snus use among females is low [10]. In recent years there have, however, been reports of increasing snus use among females in Sweden [13] and in other countries, e.g., in Norway [14]. Given these marked differences in tobacco use patterns between boys and girls, the analyses will also be stratified by gender.

It should also be noted that there are currently other forms of nicotine use that have been on the rise in recent years [15,16], such as vaping (E-cigarettes) and the use of tobacco-free snus. In Japan, these seem to have reduced the sale of ordinary cigarettes [17]. These products have, however, only been available on the Swedish market during the last couple of years, and vaping has only been experimental among youth during our study period [18]. The use of tobacco-free snus has, however, increased considerably, and the implications of this for our results are brought up in the discussion.

## 2. Materials and Methods

### 2.1. Participants

Data stem from a cross-sectional national survey conducted by the Swedish Council for Information on Alcohol and Other Drugs (CAN). CAN has conducted school surveys on alcohol and drug habits among Swedish 9th-grade students (15–16 years of age) annually since 1971. However, there have been some changes in the questions regarding tobacco use, and therefore we only used data from 1991–2020 in the present study. The survey is an anonymous paper-and-pen questionnaire and is completed in the classroom during school hours. A stratified sampling procedure in two steps was used to ensure that all regions in Sweden were represented; the school was used as the sampling unit in the first step. In the second step, one class was selected randomly in each of the selected schools. The mean response rate among the sampled schools was 85% (range: 69–96%). At the individual level (i.e., students who were present and willing to participate when the questionnaire was filled out), the mean response rate was 85% (range: 78–90%). A standardized approach was used to clean the data from incomplete or obviously exaggerated responses. About 1–2% of the respondents were excluded each year according to this standardized procedure. The sample was weighted by region and gender [19]. The total sample comprised 163,617 individuals (with a fairly equal proportion of boys and girls) representative of 9th-grade students in Sweden across the period 1991–2020 (see Table 1 for details).

### 2.2. Measures

Between 1991–2012 lifetime cigarette smoking and lifetime use of snus were measured with the questions “Do you smoke/use snus?” Those who answered “Yes”, “No, only tried”, or “No, have quit” were classified as lifetime cigarette smokers/snus users. From 2012 and onwards, the questions were “Have you ever smoked cigarettes/used snus?” with the response options “No”, “Yes, during the past 30 days”, “Yes, during the past 12 months”, or “Yes, more than 12 months ago”. Those who answered any of the yes options were classified as lifetime cigarette smokers/snus users. However, these changes do not affect the ability to analyze the long-term trends in tobacco use among Swedish adolescents [19]. In this study, we distinguished between lifetime use of cigarettes only, lifetime use of snus only, and lifetime use of both cigarettes and snus (dual users). These three measures are mutually exclusive and together make up the fourth measure of lifetime use of any of these tobacco products, which is the added sum of the three other measures.

### 2.3. Statistical Analyses

In addition to a graphical description of trends in the various measures of tobacco use, we measured the correlation between these trends with the Pearson correlation coefficient (Rxy). The significance level was set at *p* < 0.05 for all analyses.

## 3. Results

In Figure 1, the trends in tobacco use between 1991 and 2020 are depicted. Total tobacco use has decreased markedly among Swedish 9th graders over the past 30 years. In 1991, 72% of the 9th graders reported the use of any tobacco, whereas the prevalence had declined to 31% in 2017. Between 2017 and 2020, tobacco use increased, and 36% reported the use of any tobacco in 2020. A similar decline is observed for dual use of cigarettes and snus and use of cigarettes only, whereas the trend in the use of snus only is more stable.

Across almost the entire study period, the use of cigarettes was more common than the use of snus. More than 90% of tobacco users reported smoking cigarettes, either solely or both snus and cigarettes, whereas lifetime use of snus only was rare. However, since 2017 there has been a shift, and the use of snus has increased markedly. This development has brought with it that snus use is now more common among tobacco users than smoking cigarettes (see Table 2). Furthermore, the increasing use of snus underlies the increase in total tobacco use among Swedish 9th graders observed since 2017.

Figure 2 and Figure 3 show the corresponding trends stratified by sex. The trends in total tobacco use, dual use, and use of cigarettes only were similar among boys and girls and followed, thus, the overall declining trend. Boys reported a slightly higher level of tobacco use compared to girls due to a higher prevalence of use of snus, whereas girls had a higher prevalence of cigarette-only use.

Snus accounted for an increasing share of overall tobacco use across the study period among both girls and boys. In 1991, the use of snus only was reported by 7% of boys, whereas it was estimated to be 14% in 2020. Although lifetime use of snus only was much less common among girls, it increased from 1% in 1991 to 7% in 2020. Furthermore, both girls and boys have an increase in overall tobacco use after 2017 due to an increase in snus use.

The trend in the prevalence of total tobacco use was positively correlated with the trend in the prevalence of dual use (Rxy = 0.98, *p* < 0.05) and the trend in the prevalence of cigarette-only use (Rxy = 0.87, *p* < 0.05). This confirms the graphical impression that the decline in total tobacco use coincides with the decline in smoking.

The trend in the prevalence of total tobacco use was, on the other hand, negatively correlated with the prevalence of use of snus only (Rxy = −0.41, *p* < 0.05). Furthermore, the correlation between trends in the prevalence of cigarette use only and snus use only was negative (Rxy = −0.71, *p* < 0.05). These correlations were not obvious in the graphical presentation and suggested that increasing snus use may be linked to less smoking and, by that, lower tobacco use in general, at least on average for the period 1991–2020.

A similar pattern regarding the correlation between the different measures of tobacco use was found among boys and girls. The negative correlation between snus use only and overall tobacco use and smoking was, however, not statistically significant in gender-specific analyses.

Given the change observed after 2017, similar correlations were calculated for this shorter period. A positive correlation was found between total tobacco use and prevalence of snus use only, as well as dual use. The correlation between the prevalence of cigarette use only and total tobacco use was, however, negative. This verifies the graphical impression that the recent rise in snus use was related to an increase in total tobacco use among adolescents in Sweden.

## 4. Discussion

This paper aimed to examine trends in tobacco use among 9th graders in Sweden during the period 1991–2020. We found that total tobacco use has more than halved in this age group and that a decline is observed for both dual use and use of cigarettes only. The trend in the use of snus only is more stable and also displays a sharp increase since 2017. Cigarettes used to be much more common than snus, but the development in recent years has brought with it that snus use in 2020 has become more common than smoking cigarettes in this age group. These overall trends apply to both girls and boys, although cigarette use is still somewhat more common among girls, and snus use is clearly more prevalent among boys.

During the entire period, there was a positive correlation between total tobacco use and the prevalence of both dual use and the use of cigarettes only. This suggests that the decline in total tobacco is the result of a decline in cigarette use; both cigarettes only and dual use, where cigarettes are used along with snus. Furthermore, the correlation between trends in the prevalence of cigarette use only and snus use only was negative, suggesting that increasing snus use may have been linked to less smoking and, therefore, also indirectly to lower overall tobacco use, at least up to 2017.

A thought-provoking finding was that the situation seems to have changed after 2017 when an increasing prevalence of snus use is related to increased tobacco use despite declining smoking figures. The increase in snus use is evident both in terms of snus use only and in dual use. This is especially the case among girls, where the prevalence of snus use among tobacco users has increased by 76 percent between 2017 and 2020. The introduction of “All white snus” in 2014 on the Swedish market is a possible explanation for this increase. This form of snus distinguishes itself from traditional snus in the way that it does not contain tobacco and is, therefore, not regulated by Swedish law. Since the survey question about the use of snus does not discriminate between different forms of snus, it is likely that the rise in snus use found after 2017 is due to increasing reports of tobacco-free snus. It is worth noting that increasing snus use among girls is also reported in Norway [20].

From a public health perspective, the trends of declining tobacco use, and especially cigarette use, are positive since tobacco is one of the leading contributors to the disease burden [1]. One possible conclusion from our findings is that introducing snus would be a good way to reduce cigarette use. However, with cigarette use also declining in countries other than Sweden (such as in the United States [21]), we believe that other studies are needed to definitively say that this is the case (as snus is not prevalent in the US). The evidence for the health benefits of snus is also not so clear that we would be comfortable with saying that this is preferable to cigarette use.

Furthermore, the increase in recent years in dual use of cigarettes and snus warrants some caution as dual use has been found to increase both the severity of addiction [20] and the likelihood of continuing to use nicotine [8]. There are also some indications of more severe problems among those calling the national helpline for smoking cessation [20], which in turn require more directed and tailored help.

During the past two decades, there has also been a large decline in alcohol use among Swedish adolescents [22,23] and internationally [24,25]. Involvement in criminal behaviour has also declined among youth in Sweden during this period [26]. There are also reports of declines in risk behaviors in general among adolescents [27]. The trends of declining tobacco use up to 2017 might, therefore, also be seen as part of a larger shift towards young people today being more well-behaved and health-conscious.

Previous studies have also shown that tobacco consumption and smoking behavior are linked to macroeconomic fluctuations and events in the adult population [28,29]. With the present study covering over 30 years, the study period includes both economic crises and periods of economic growth. No clear pattern is, however, apparent that would indicate that youth tobacco use in Sweden would be associated with macroeconomic shifts, and instead, the trends are progressing rather continuously over time, most notably the decline in overall tobacco use.

This study is based on repeated cross-sectional surveys and self-reported data, and this needs to be kept in mind when interpreting the results. The cross-sectional nature of the data means that we do not know which behavior preceded the other, i.e., we do not know if the adolescents started using cigarettes or started using snus. Furthermore, we do not know if the correlations found here are causal or due to some other factor, for instance, a general trend towards a healthier lifestyle among adolescents. Self-reported data usually also entails some under-reporting [30]. This kind of data has, however, shown good reliability when it comes to alcohol use among adolescents [31], and our focus on changes over time should make these issues less salient. There were also some changes to how the questions were posed during the study period, which limits the possibility of comparing levels of use over time.

A socioeconomic patterning of tobacco use has been observed in previous studies, with smoking and dual use of both snus and cigarettes more prevalent among those with low income and low education, whilst snus use is more common among those with intermediate education [7]. It is possible that across the entire study period of the present study, there might have been changes in the socioeconomic composition of the population and that this would then have affected the prevalence rates. However, data on socioeconomic status was not available, and we were not able to elaborate on this question. Future studies should examine how the socioeconomic composition of the population is associated with both the use of snus and cigarettes and how this interacts and forms trends in tobacco use among adolescents in Sweden.

The major strength of the current study is the high-quality data at hand, with data being collected by the same organization using the same mode of collection during the entire period. The large nationally representative samples also mean that the results are highly generalizable to Swedish adolescents in general and not subject to regional variations or trends in certain sub-groups. The large samples also lower the risk of the results being produced by random variations. Having data covering a 30-year period with questions on both the use of cigarettes and snus allowed us to give a detailed and thorough examination of the trends and development in tobacco use among 9th graders from Sweden and also allowed us to examine how the use of cigarettes and snus interact and shape the overall trend in tobacco use.

## 5. Conclusions

The sharp decline in tobacco use among 9th graders in Sweden over the past three decades is driven by declining cigarette use. The correlations between the various forms of tobacco use suggest that snus use may have contributed to the decline in cigarette use and, by that, in overall tobacco use. The situation changed after 2017 when a sharp rise in snus use seems to have increased total tobacco use among adolescents in Sweden. More research is needed to draw more specific implications on policy on the basis of these findings, which are very contextually dependent on Sweden, where snus has been used for a very long time.

## Figures and Tables

**Figure 1 ijerph-20-05262-f001:**
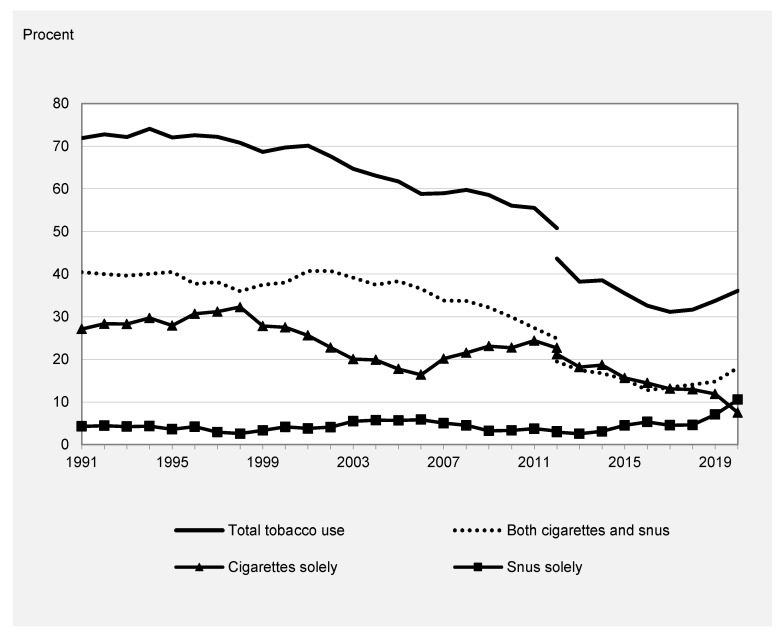
Use of tobacco among 9th graders in Sweden, 1991–2020.

**Figure 2 ijerph-20-05262-f002:**
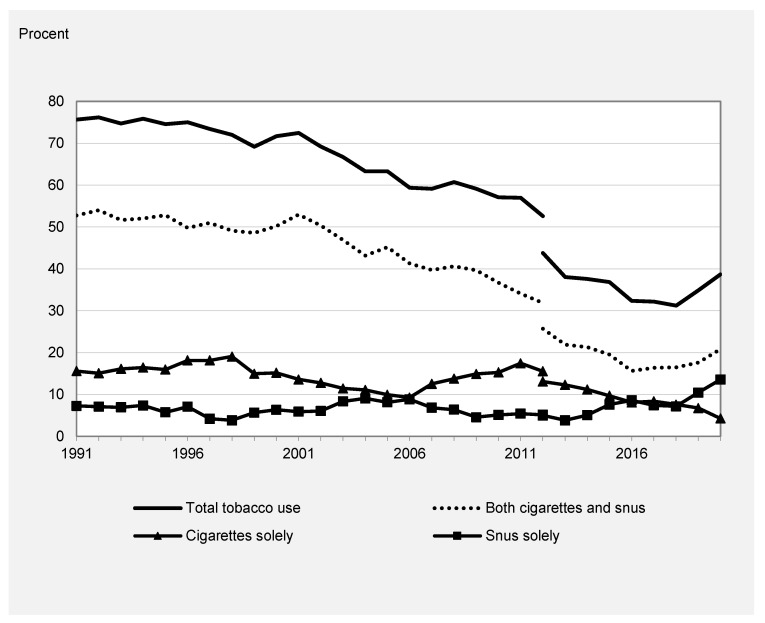
Use of tobacco among 15–16-year-old males in Sweden, 1991–2020.

**Figure 3 ijerph-20-05262-f003:**
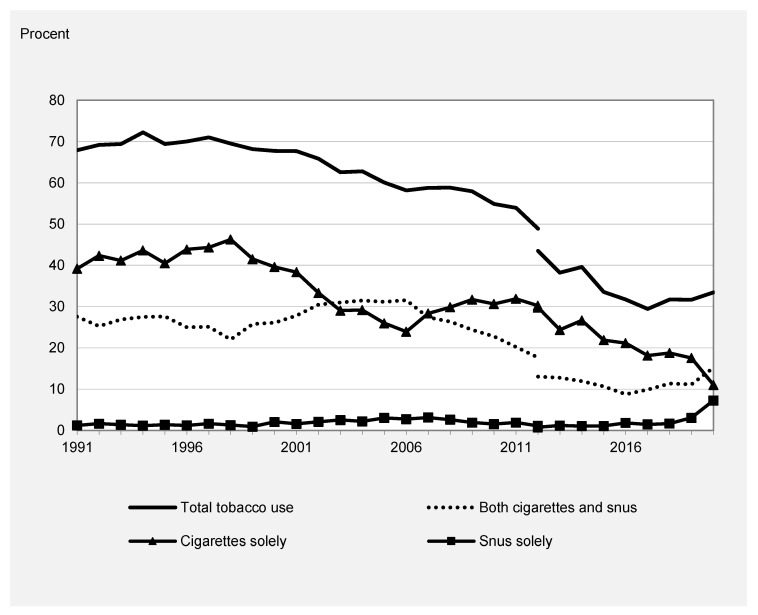
Use of tobacco among 15–16-year-old females in Sweden, 1991–2020.

**Table 1 ijerph-20-05262-t001:** Descriptive statistics.

	Number of Respondents	Response Rate (%)	Proportion Girls
1991	5865	86	49.4
1992	5829	87	48.8
1993	5903	90	48.8
1994	5847	90	50.4
1995	5575	88	49.3
1996	6016	89	49.1
1997	5674	89	48.4
1998	5442	87	49.5
1999	5182	86	48.5
2000	5334	86	50.5
2001	5474	85	49.9
2002	5456	85	48.8
2003	5264	86	48.8
2004	5420	85	50.9
2005	5423	85	49.8
2006	4946	85	48.9
2007	5319	83	48.0
2008	4895	84	49.1
2009	5219	85	50.0
2010	4851	84	51.6
2011	4655	83	49.5
2012a	4545	84	49.7
2012b	4889	84	50.3
2013	5109	85	48.7
2014	4932	85	47.5
2015	4961	85	49.1
2016	4805	83	48.8
2017	6124	83	48.1
2018	5319	83	49.1
2019	5214	83	49.1
2020	4130	78	47.4

**Table 2 ijerph-20-05262-t002:** The proportion of smokers and snus users among tobacco users, 1991–2020.

	Total		Males		Females	
	Proportion of Smokers among Tobacco Users	Proportion of Snus Users among Tobacco Users	Proportion of Smokers among Tobacco Users	Proportion of Snus Users among Tobacco Users	Proportion of Smokers among Tobacco Users	Proportion of Snus Users among Tobacco Users
1991	94	62	90	79	98	42
1992	94	61	91	80	98	39
1993	94	61	91	78	98	41
1994	94	60	90	78	98	40
1995	95	61	92	79	98	42
1996	94	58	91	76	98	37
1997	96	58	94	77	98	38
1998	96	54	95	74	98	34
1999	95	59	92	78	99	39
2000	94	61	91	79	97	41
2001	95	63	92	81	98	43
2002	94	66	91	82	97	49
2003	92	69	87	83	96	54
2004	91	69	86	82	97	54
2005	91	71	87	84	95	57
2006	90	72	85	84	95	59
2007	91	66	88	79	95	52
2008	92	64	90	77	96	49
2009	94	61	92	75	97	45
2010	94	59	91	73	97	44
2011	93	56	90	69	97	41
2012	93	51	89	70	98	32
2013	93	52	90	68	97	36
2014	92	52	87	70	97	33
2015	87	56	79	74	97	35
2016	84	56	73	75	94	33
2017	85	58	77	74	95	38
2018	85	59	77	76	95	41
2019	79	65	70	81	90	45
2020	71	79	65	89	78	67

## Data Availability

The data presented in this study are available upon reasonable request from the corresponding author. The data are not publicly available due to privacy and ethical restrictions.

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
