# Peer review of "Trends in Tobacco Use among 9th Graders in Sweden, 1991–2020"

_ijerph, 2023, doi:10.3390/ijerph20075262_

Round 1

Reviewer 1 Report (Previous Reviewer 1)

The authors have responded correctly to the comments that I made for the minor revision.

Author Response

Response: Thank you.

Reviewer 2 Report (New Reviewer)

Author Response

  1. Final Draft? -- The draft seems to not be a final draft that one would submit to a journal. It seems to be an edited draft with the edits still in the paper. And it raises issues about why they did the edits they did. So, a reviewer is not just reviewing the paper but the appropriateness of their edits. They should have submitted a non-edited complete draft.

Response: This is a revised version of a manuscript previously submitted to this journal. We choose to keep the track changes as this would make it easier for (the previous) reviewers and editors to follow the changes we made as per their suggestions. We are sorry that the journal did not make this clear to you.

  1. Statistics -- Correlations are OK to use, but there may be other statistics to consider in examining trends. It might be good to further justify the statistics chosen.

Response: We have opted for correlations since we think this is the best option for the current manuscript. We would gladly listen to other concrete suggestions, but will keep the analyses as they are for the moment, since you also note that these are OK.

  1. More could have been done from these data -- The data did not seem to be analyzed to the extent that they could be. While it is not clear exactly what variables they have, it would be important to know if these trends varied by some key variables, socio-economic status or ethnicity (if there is much variance in ethnicity from immigration). Or if these data were not collected, it should be noted. Is there any data on why females are less likely to use snus?

Response: Neither ethnicity nor socioeconomic status is available. We do not think it is feasible to list all variables not included as this list would be very long. Snus use is (has been) a heavily gendered practice in Sweden, but we have no data to examine why this is the case.

  1. Public Health Implications – There are major public health implication to these data that are not drawn out. What are the public health implication of these trends? In some ways, I think that these data are more important than the authors. What do these data mean for policy? What are they adding to global knowledge that makes a difference? The conclusion seem tepid given the significance of their data.

Overall, I think that this is an important paper that can be further developed and make a more significant contribution.

Response: Thank you for these kind words. We do agree that we are cautious in our conclusions about the implications of our results. We do however think that there is good reason to be cautious. One implication of our results could be that introducing snus is a good way to reduce cigarette use. With cigarette use declining also in countries other than Sweden (like the US) we are not comfortable with definitively saying that this is the case (as snus is not prevalent in the US). The evidence for the health benefits of snus is also not so clear that we would be comfortable with saying that this is preferable to cigarette use. Since there is also evidence of dual users being worse off the introduction of snus to a new market could also have detrimental effects, so our results are very context dependent for Sweden where snus has been used for a very long time.

Reviewer 3 Report (New Reviewer)

- Abstract part is more than an abstract, it is an expression of the problem. The authors need to report clearly the objective, method, result, and conclusion.

- Method: Each of measures should be carefully presented with the validity and reliability index.

- The procedure of the study has not been reported.

Author Response

Comments and Suggestions for Authors

- Abstract part is more than an abstract, it is an expression of the problem. The authors need to report clearly the objective, method, result, and conclusion.

Response: Apologies but we do not agree with the reviewer and will keep the abstract as is. Please give detailed information on what you think the abstract is missing and we will happily add that information.

- Method: Each of measures should be carefully presented with the validity and reliability index.

Response: These are single item questions so there is no additional statistics (like for example cronbach’s alpha) to report.

- The procedure of the study has not been reported.

Response: Please give details as to where the description of the procedure is lacking and we will try to address this.

Round 2

Reviewer 3 Report (New Reviewer)

I don't have any comments since the authors modified the manuscript as indicated. Thank you. 

This manuscript is a resubmission of an earlier submission. The following is a list of the peer review reports and author responses from that submission.

Round 1

Reviewer 1 Report

CENTRAL AND GENERAL ISSUES

Summary

This paper examines trends in tobacco use among Swedish ninth grade students over the period 1991-2020. Using annual school surveys with nationally representative samples, the correlation between these trends was calculated using the Pearson correlation coefficient (Rxy). The results suggest that the sharp decline in tobacco use among ninth graders in Sweden over the past three decades is due to the decline in cigarette smoking. I believe that the result adds value to the literature on the analysis of the tobacco market. However, I think there are some important aspects that need to be improved before recommending its publication in International Journal of Environmental Research and Public Health.

Specific Comments

1. The introductory section does not explicitly talk about how tobacco companies are using electronic devices and heated tobacco to replace traditional cigarettes with these new alternatives. A recent article shows how Philip Morris International is using heated tobacco to replace the traditional cigarette. In this line, I think the authors should reflect on this and cite the following papers:

Stoklosa, M., Cahn, Z., Liber, A., Nargis, N., & Drope, J. (2020). Effect of IQOS introduction on cigarette sales: evidence of decline and replacement. Tobacco Control, 29(4), 381-387. http://dx.doi.org/10.1136/tobaccocontrol-2019-054998

Golpe, A. A., Martín-Álvarez, J. M., Galiano, A., & Asensio, E. (2022). Effect of IQOS introduction on Philip Morris International cigarette sales in Spain: a Logarithmic Mean Divisa Index decomposition approach. Gaceta Sanitaria. 36 (4), 293-300. https://doi.org/10.1016/j.gaceta.2021.12.007

2. Another relevant issue that I think should be taken into account in the paper is the effect of the business cycle on cigarette consumption. The period analyzed is long, so the authors should take this into account and cite the following papers:

Bellés-Obrero, C., & Castello, J. V. (2018). The business cycle and health. In Oxford Research Encyclopedia of Economics and Finance. https://doi.org/10.1093/acrefore/9780190625979.013.282

Martín-Álvarez, J. M., Almeida, A., Galiano, A., & Golpe, A. A. (2020). Asymmetric behavior of tobacco consumption in Spain across the business cycle: a long-term regional analysis. International Journal of Health Economics and Management, 20(4), 391-421. https://doi.org/10.1007/s10754-020-09286-y

3. In the conclusions I do not see any paragraph showing the limitations of this work. It would be important for the limitations of this paper to be made clear.

Author Response

  1. The introductory section does not explicitly talk about how tobacco companies are using electronic devices and heated tobacco to replace traditional cigarettes with these new alternatives. A recent article shows how Philip Morris International is using heated tobacco to replace the traditional cigarette. In this line, I think the authors should reflect on this and cite the following papers:

Stoklosa, M., Cahn, Z., Liber, A., Nargis, N., & Drope, J. (2020). Effect of IQOS introduction on cigarette sales: evidence of decline and replacement. Tobacco Control, 29(4), 381-387. http://dx.doi.org/10.1136/tobaccocontrol-2019-054998

Golpe, A. A., Martín-Álvarez, J. M., Galiano, A., & Asensio, E. (2022). Effect of IQOS introduction on Philip Morris International cigarette sales in Spain: a Logarithmic Mean Divisa Index decomposition approach. Gaceta Sanitaria. 36 (4), 293-300. https://doi.org/10.1016/j.gaceta.2021.12.007

Response: Thank you for pointing out these two interesting papers. We do however fail to see how they are relevant for the current manuscript as this does not pertain to electronic and heated tobacco products. These newer types of products have also only been on the market during the last couple of years and can therefore only be relevant for a few years towards the end of the studied period. With our study covering a 30 year period the possible importance of this is less salient.

  1. Another relevant issue that I think should be taken into account in the paper is the effect of the business cycle on cigarette consumption. The period analyzed is long, so the authors should take this into account and cite the following papers:

Bellés-Obrero, C., & Castello, J. V. (2018). The business cycle and health. In Oxford Research Encyclopedia of Economics and Finance. https://doi.org/10.1093/acrefore/9780190625979.013.282

Martín-Álvarez, J. M., Almeida, A., Galiano, A., & Golpe, A. A. (2020). Asymmetric behavior of tobacco consumption in Spain across the business cycle: a long-term regional analysis. International Journal of Health Economics and Management, 20(4), 391-421. https://doi.org/10.1007/s10754-020-09286-y

Response: Thank you. Again, we do not see how the content of these papers would fit in the current manuscript.

  1. In the conclusions I do not see any paragraph showing the limitations of this work. It would be important for the limitations of this paper to be made clear.

Response: We agree and raise limitations on page 7, lines 191-201.

Submission Date

17 November 2022

Date of this review

29 Nov 2022 19:25:53

Open Review

English language and style

( ) English very difficult to understand/incomprehensible
( ) Extensive editing of English language and style required
( ) Moderate English changes required
(x) English language and style are fine/minor spell check required
( ) I don't feel qualified to judge about the English language and style

Yes

Can be improved

Must be improved

Not applicable

Does the introduction provide sufficient background and include all relevant references?

( )

( )

(x)

( )

Are all the cited references relevant to the research?

( )

(x)

( )

( )

Is the research design appropriate?

( )

( )

(x)

( )

Are the methods adequately described?

( )

( )

(x)

( )

Are the results clearly presented?

( )

( )

(x)

( )

Are the conclusions supported by the results?

( )

( )

(x)

( )

Comments and Suggestions for Authors

Dear authors, 

===========================================

Overall comments

This paper endeavors to examine trends in tobacco use among Swedish 9th graders over the period 1991-2020 with a total sample of 163 617 students. The authors have indicated, as the main finding, is the high-quality of the data.

1 Merits

The paper itself is well written, although somewhat descriptive. The authors have undertaken an interesting and rigorous piece of data collection.

2 Critics

  1. Firstly, the work is mostly descriptive and not focused on specific hypotheses, the statistical analysis were not treated rigorously.

Response: We agree that the paper is descriptive in nature, we were however very rigorous in our statistical analyses and believe the findings of the paper are of interest to the research community.

  1. While I found the title, the topic and the data of the paper appealing, I must confess I found that the main values of this contribution are hidden (for me) and the approach is over-simplified. I agree with the authors that the strength of the current study is the high-quality of the used data. But I still convinced that either send this article to a journal that deals exclusively with data valuation and highlight the quality of these data in that journal, or analyze these data with much more statistical rigor. In this second case, the contribution will be double.

Response: We do not share the opinion of the reviewer and think that the manuscript and the findings make an contribution to the literature on tobacco consumption among youth.

  1. The data analysis steps should be detailed.

Response: Please be specific so that we can clarify any steps.

  1. Since the authors focus on simple correlations of some trends. It is potentially desirable to use some statistical tests  (differences between the observed groups) such as classical tests of Independence?.

Response: We are happy to provide additional tests, we do however not follow what the reviewer wants tested. Please be more specific.

  1. On correlation analysis, eg. Lines 132–>135 «The trend in prevalence of total tobacco use was positively correlated with the trend 132 in prevalence of dual use (Rxy=0,98***) and the trend in prevalence of cigarettes only use 133 (Rxy=0,87**
    1. first, what do you mean by stars ** (statistical significance ?)

Response: Yes, this is correct. We have now replaced the starts with p<0.05 instead to avoid confusion.

  1. it is normal to find a correlation between Total tabacco use and double use or cigarettes solely, since the Total variable is constructed using both sub-variables (cigarettes solely, ... ). There is always correlation between X and Y when Y=X1+X2 .

Response: In this paper we use three variables of tobacco use, snus only, cigarettes only and dual use. These make up the total tobacco use. So there is not always a correlation between the X and Y as the reviewer suggests as we do not find a correlation between snus and total tobacco. This is why we state that “that the decline in total tobacco use coincides with the decline in smoking”. If we also would have found a correlation between total and snus use then the decline would not be the result of a decline in smoking.

  1. In Figures : the dual use = snus +cigarette smoking. What about the total use ?

Response: In the figures (and the entire manuscript) total use equals dual use+snus+cigarettes.

  1. with Rxy =-0.41, the correlation is very weak between variables. It is unclear for why the authors say «... correlated negatively..»

Response: The correlation was statistically significant. We leave it up to the reader to judge if it is a weak or strong correlation.

  1. The correlation analyses lack rigor. Hypotheses about the prevalence of total tobacco use should be provided with other variables (observable or not, such as socioeconnomic factors) and then linked them to the trend decline.

Response: Apologies, but we do not understand this comment.

  1. The results section should clearly explain the key limitations of prior work that are relevant to this paper.

Response: We don’t think the result section should include previous work but rather be dedicated to the results of the current manuscript. Previous work and their limitations are discussed in the intro and discussion sections of the manuscrip

Reviewer 2 Report

Dear authors, 

===========================================

Overall comments

This paper endeavors to examine trends in tobacco use among Swedish 9th graders over the period 1991-2020 with a total sample of 163 617 students. The authors have indicated, as the main finding, is the high-quality of the data.

Merits

The paper itself is well written, although somewhat descriptive. The authors have undertaken an interesting and rigorous piece of data collection.

Critics

  1. Firstly, the work is mostly descriptive and not focused on specific hypotheses, the statistical analysis were not treated rigorously.
  2. While I found the title, the topic and the data of the paper appealing, I must confess I found that the main values of this contribution are hidden (for me) and the approach is over-simplified. I agree with the authors that the strength of the current study is the high-quality of the used data. But I still convinced that either send this article to a journal that deals exclusively with data valuation and highlight the quality of these data in that journal, or analyze these data with much more statistical rigor. In this second case, the contribution will be double.
  3. The data analysis steps should be detailed.
  4. Since the authors focus on simple correlations of some trends. It is potentially desirable to use some statistical tests  (differences between the observed groups) such as classical tests of Independence?.
  5. On correlation analysis, eg. Lines 132–>135 «The trend in prevalence of total tobacco use was positively correlated with the trend 132 in prevalence of dual use (Rxy=0,98***) and the trend in prevalence of cigarettes only use 133 (Rxy=0,87**
    1. first, what do you mean by stars ** (statistical significance ?)
    2. it is normal to find a correlation between Total tabacco use and double use or cigarettes solely, since the Total variable is constructed using both sub-variables (cigarettes solely, ... ). There is always correlation between X and Y when Y=X1+X2 .
      1. In Figures : the dual use = snus +cigarette smoking. What about the total use ?
      2. with Rxy =-0.41, the correlation is very weak between variables. It is unclear for why the authors say «... correlated negatively..»
    3. The correlation analyses lack rigor. Hypotheses about the prevalence of total tobacco use should be provided with other variables (observable or not, such as socioeconnomic factors) and then linked them to the trend decline.
  6. The results section should clearly explain the key limitations of prior work that are relevant to this paper.

Author Response

(The authors gave the same response as above.)
